# Application of type II diabetes incidence and mortality rates for insurance

**Jack C. Yue[1], Hsin-Chung Wang[2]\*, Ting-Chung Chang[3]**

**1** Department of Statistics, National Chengchi University, Taipei, Taiwan, Republic of China, **2** Department of Statistical Information and Actuarial Science, Aletheia University, New Taipei City, Taiwan, Republic of China, **3** Department of Accounting Information, Chihlee University of Technology, New Taipei City, Taiwan, Republic of China

\* au4369@mail.au.edu.tw

## Abstract

Prolonging life is a global trend, and more medical expenditure is being spent on chronic diseases owing to population aging. Diseases commonly seen in middle-aged and elderly people, such as heart disease and diabetes, have slowed mortality improvement in recent years. Diabetes is a common chronic disease and comorbidity of many serious health conditions. The total estimated cost of diabetes in the United States was $327 billion in 2017. However, many people are unaware that diabetes is common, and at least 21.4% of adults do not know that they have diabetes. The number of diabetes-related deaths has been increasing, and diabetes was the 5th cause of death in Taiwan in 2019. In this study, we explore the trend and influence of diabetes in Taiwan and apply mortality models, such as the Lee-Carter and Age-Period-Cohort models, using data from Taiwan's National Insurance to model the incidence and mortality rates of diabetes. We found that the Lee-Carter model provides fairly satisfactory estimates and that people with diabetes regularly taking diabetes medication have lower mortality rates. Moreover, we demonstrate how these results can be used to design diabetes related insurance products and prepare the insured to face the impact of incurring diabetes. In addition, we consider different criteria for judging whether people have diabetes (as there is no consensus on these criteria) and investigate the issue of moral hazard in designing diabetes insurance products.

**Data Availability Statement:** The data underlying this study is from the National Health Insurance Research Database (NHIRD), which has been transferred to the Health and Welfare Data Science Center (HWDC). Interested researchers can obtain

## Section 1: Introduction

Extending lifespan is a global trend in the 21st century, and population aging is becoming more apparent in many countries. The increase in life expectancy is noticeable in many Asian countries, although it is slowing in many developed countries (Fig 1). As a result of prolonged life, people are paying more attention to retirement planning, including economic, medical, and long-term care needs [1]. This study focuses on the medical needs of the elderly population. Elderly individuals generally have higher medical utilization for inpatient and outpatient visits. For example, the proportion of elderly in Taiwan was approximately 14.6% in 2018, but

the data through formal application to the HWDC, Department of Statistics, Ministry of Health and Welfare, Taiwan: https://dep.mohw.gov.tw/DOS/cp-5283-63826-113.html.

**Funding:** The author(s) received no specific funding for this work.

**Competing interests:** The authors have declared that no competing interests exist.

their medical expenditure in national health insurance was over 38.2% (Source: Ministry of Health and Welfare, Taiwan).

The higher medical utilization of elderly people in Taiwan is often associated with their chronic conditions. For example, approximately 3/4 and 1/2 of Taiwan's elderly population have at least one and two chronic diseases, respectively. The proportion of deaths related to metabolic syndromes, such as heart disease, stroke, and type 2 diabetes mellitus (T2DM), has become the leading cause of death in Taiwan, surpassing that related to cancer (which is still the single cause of death). Among these diseases, diabetes is often overlooked and does not receive as much attention as heart disease and stroke. However, people have gradually realized its impact on health [2–4]. The global prevalence of diabetes increased from 4.7% to 8.5% between 1980 and 2014 (World Health Organization (2016). Several factors contribute to accelerated diabetes epidemic and, for example, poor diabetes management puts people into higher risk of serious complications [5–9]. In general, diabetes affects Europeans in developed countries when individuals are 65 years and older [10], whereas people from South Asia are more likely to have T2DM than people from other continents or countries [11].

Approximately 1.5 million worldwide had died of diabetes in 2012, this number could be higher considering that diabetes increases the risk of death for those with other health conditions. In Taiwan, cancer remains the focus of people's health concern, and cancer patients with diabetes have a 40–80% higher risk of death than those without diabetes [12]. Additionally, diabetes was recently identified as a significant risk factor for death among COVID-19 patients [13]. Many causes of death (e.g., kidney and cardiovascular diseases) are highly correlated with

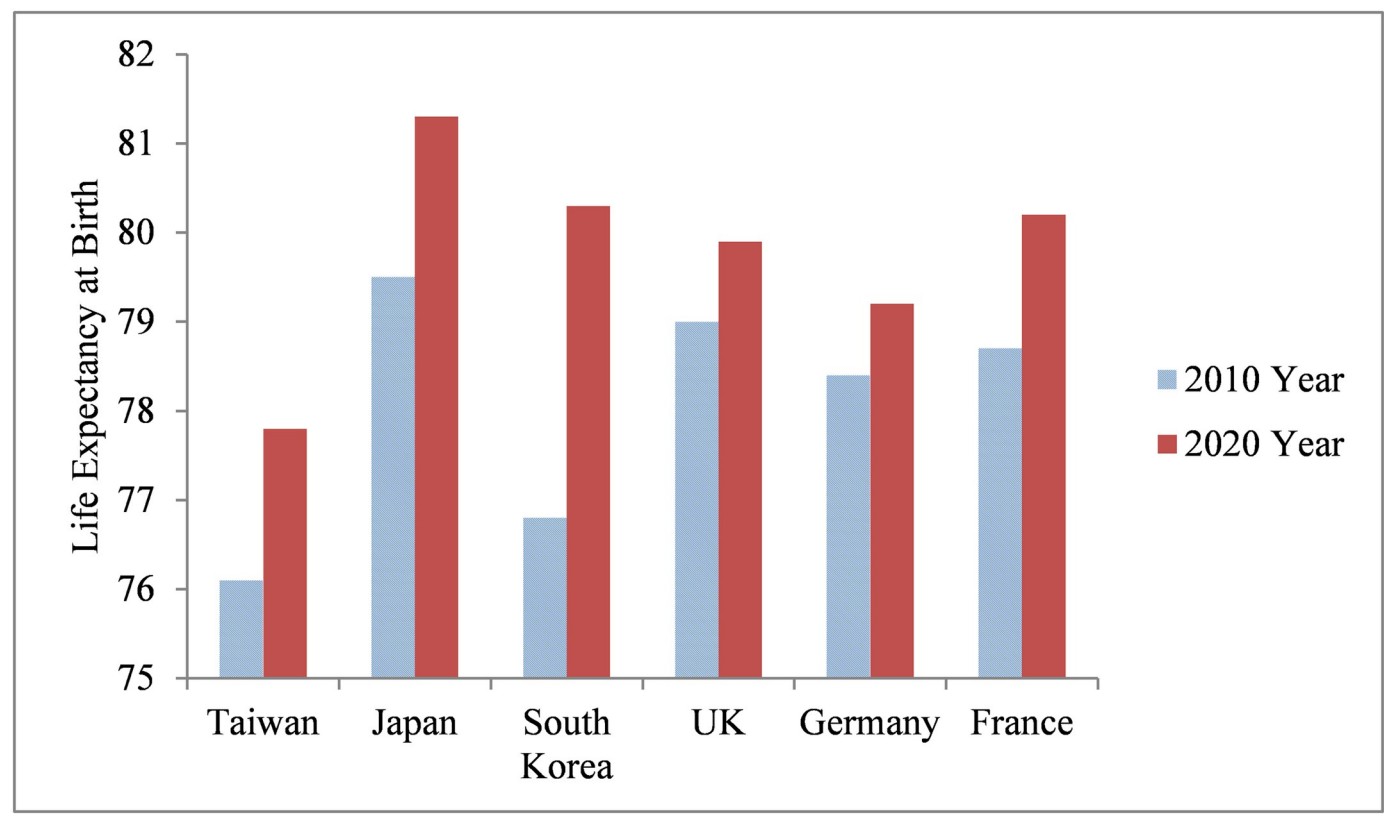

**Fig 1. Male life expectancy at birth for selected countries.** Source: National Development Council, Taiwan, R.O.C.

diabetes, although it might not be a direct cause. Moreover, medical expenditures related to diabetes have become more noticeable, and are likely to increase in many countries. For example, the direct expenses for diabetes were over $82.7 billion in 2012. According to the International Diabetes Federation (IDF) Diabetes Atlas 2019 [14], the total healthcare expenditure on diabetes was estimated at $760 billion in 2019 and is expected to reach $845 billion by 2045.

The influence of diabetes on health and life expectancy is predicted to increase, and the insurance industry can play an important role in managing the consequences. This study explores the feasibility of designing diabetes-related insurance products based on datasets from Taiwan's National Health Insurance Research Databases (NHIRD). The datasets we selected covered nearly 50% of Taiwan's population aged over 50, a scale not seen in previous studies. All hospital visits, including inpatient, outpatient, and surgical records, were assessed using those datasets. Diabetes is generally associated with a higher mortality rate in Taiwan, and individuals with diabetes find it difficult to purchase life insurance. We study the mortality rates of patients with diabetes and investigate whether insurance companies can cover these patients.

However, unlike cancer and catastrophic illness (CI) [15], the definition of diabetes has been controversial. The government, insurance industry, and doctors have different opinions on diabetes, which creates difficulties in designing diabetes insurance products. Many criteria for defining diabetes have been proposed in previous studies, many of which include diabetes clinic visits as a necessary condition (S1 Table) [16–21]. In other words, the incidence rates of diabetes (and possibly the mortality rates of patients with diabetes and their medical utilization) depend on the criteria used, and it is difficult to determine the most appropriate criteria for designing insurance products. Instead, we compare different criteria for judging T2DM and choose those that can provide stable and consistent estimates of incidence and mortality rates.

The remainder of this paper is organized as follows. We briefly introduce the datasets used in this study, and describe how we define T2DM and the mortality models used in Section 2. Empirical data analysis, including the processes of data cleaning and big data analysis, is presented in Section 3. The Modelling and application of diabetes incidence and mortality rates to design insurance products are presented in Section 4. The final section presents the discussion of our findings.

## Section 2: Materials and methods

The NHIRD is an important public resource and has been used in many research studies for more than 20 years as almost the entire Taiwanese population is enrolled in it. The research topics related to NHIRD include, for example, the hospital utilization and medical usage of cancer patients in Taiwan [22–24]. In this study, we chose two NHIRD sample datasets, the Longitudinal Health Insurance Database 2005 (LHID2005) and Elderly Longitudinal Health Insurance Database 2005 (ELHID2005), and used these datasets to acquire diabetes incidence rates, mortality rates, and medical utilization of patients with diabetes. The datasets contained one million randomly selected people who were alive in 2005, and their medical records between 1996 and 2013. The medical records included "registry for beneficiaries" (personal identification, or ID file), "ambulatory care expenditures by visits" (outpatient visit or CD file), and "inpatient expenditures by admissions" (DD file).

The major difference between the two datasets was the age range of the samples: ages 0–99 for LHID2005 and ages 65–99 for ELHID2005, and the samples selected (one million people) accounted for 4.6% and 45.7% of Taiwan's population at ages 0–99 and 65–99, respectively. Notably, the age-specific prevalence rates of T2DM increased with age, especially in the elderly

[10, 25–28] and the mortality rates in elderly patients with diabetes were also higher [29]. Therefore, we focused on people aged 45–99 and chose a sampling ratio of 45.7% for the elderly (ELHID2005) to provide stable estimates for those aged 65 and above. Moreover, the data quality (including data format and data collection) of the NHIRD has been improving since Taiwan started the NHI in 1995, thus, we used data for 2003–2013 to ensure the credibility of our analysis.

In particular, we applied frequently used mortality models to estimate the incidence and mortality rates and to determine which disease criteria produce smaller estimation errors. We used the mean absolute percentage error (MAPE) to evaluate the different disease criteria and mortality models. The MAPE is defined as

$$MAPE = \frac{1}{n}\sum_{i=1}^{n}\frac{|Y_i - \hat{Y}_i|}{Y_i} \times 100\%,$$

where $Y_i$ and $\hat{Y}_i$ are the observed and estimated values of observation $i$, $i = 1,2,\ldots,n$. According to Lewis [30], predictions with MAPE less than 10% and greater than 50% are considered highly accurate and unacceptable, respectively. The two datasets from the NHIRD (particularly the ELHID2005) were used to verify the disease criteria for diabetes.

Note that we could not obtain the mortality rates of patients with diabetes from the NHI data directly because we could not access the Cause of Death Dataset from the Department of Health and Welfare when we applied to NHIRD. Nevertheless, we obtained reliable estimates of mortality rates from medical records. We adopted the criteria used in previous studies and judged whether a patient had died (e.g., Yue et al. [15] and Chen et al. [31]). The criteria for death can be applied to people with heavy medical utilization, such as patients with CI and older people (aged 50 and over). For example, the average number of outpatient visits and medical costs for patients with CI are approximately three and seven times the national average (2019), respectively. Most criteria are based on whether those individuals stop visiting doctors and usually provide fairly accurate estimates of mortality rates. We only applied the death criteria for people aged 65 and above in this study because the mortality estimates of the elderly were very close to official statistics [15].

The incidence and mortality estimates can have many fluctuations for higher age groups owing to population size, and we introduced graduation methods to smooth age-specific rates. Two smoothing techniques were used: Partial Standardized Mortality Ratio (PSMR) and stochastic mortality models. PSMR [32] is a modification of the Standardized Mortality Ratio (SMR), which was originally designed to smooth the mortality rates of small populations using mortality information (with respect to SMR) from a large (reference) population. The SMR is often used in epidemiology to compare populations with different age structures and is defined as

$$SMR = \frac{\sum_x d_x}{\sum_x e_x} \tag{1}$$

where $d_x$ and $e_x$ are the observed and expected number of deaths for age $x$, respectively. If the SMR is greater than 1, this indicates that the small population has higher overall mortality rates than the reference population. Similarly, an SMR of < 1 indicates a lower mortality rate. Thus, SMR can be treated as a mortality index. Wang et al. [33] showed that the partial SMR can be used to stabilize estimates from stochastic models.

For the partial SMR, the graduated rates satisfy

$$v_x = u_x^* \times \exp\left(\frac{d_x \times \hat{h}^2 \times \log(d_x/e_x) + (1 - d_x/\sum d_x) \times \log(\text{SMR})}{d_x \times \hat{h}^2 + (1 - d_x/\sum d_x)}\right) \tag{2}$$

or the weighted average between raw mortality rates and SMR, where $\hat{h}^2$ is the estimate of parameter $h^2$ for measuring the heterogeneity (in mortality rates) between the small area and reference populations. To avoid unreasonable results, Lee [32] suggests larger $\hat{h}^2$ values for mortality heterogeneity between different ages. When the number of deaths is smaller, there will be larger weight from the reference population to provide smoother graduated mortality rates, and the graduated value equals $\text{SMR} \times u_x^*$ when the number of deaths is zero.

$$\hat{h}^2 = \max\left(\frac{\sum\left((d_x - e_x \times SMR)^2 - \sum d_x\right)}{SMR^2 \times \sum e_x^2}, 0\right) \tag{3}$$

Mortality models can be treated as a group of graduation methods. For example, the Gompertz model is frequently used to assess the mortality rates among the elderly. In particular, we used the Generalized Age-Period-Cohort (GAPC) model [34] to fit the incidence rates and mortality trends in patients with diabetes. We considered several stochastic mortality models in this study, including the popular Lee-Carter model [35], which is a special case of the GAPC model. In addition to applying mortality models, we discussed the spillover effects of diabetes by, for example, considering the morbidity rates of ailments related to diabetes, as it is believed that diabetes is associated with many metabolic syndrome diseases.

1. Lee-Carter (LC) model:
   If $m_{xt}$ denotes the central death rate or incidence rate for a person aged $x$ at time $t$. The LC model assumes that

   $$\log(m_{xt}) = \beta_x^{(1)} + \beta_x^{(2)}\kappa_t^{(2)} + \varepsilon_{x,t}, \tag{4}$$

   with $\sum_x \beta_x^{(2)} = 1$ and $\sum_t \kappa_t^{(2)} = 0$. $\beta_x^{(i)}$ are age related parameters ($i = 1, 2$), and $\kappa_t^{(2)}$ represents the time related parameter. Note that $\beta_x^{(1)}$ is the general mortality level, and $\beta_x^{(2)}$ is the mortality improvement rate at age x, and $\kappa_t^{(2)}$ is a linear function of time. The term $\varepsilon_{x,t}$ denotes the error term and is assumed to be white noise with zero mean and a relatively small variance.

2. Renshaw-Haberman (RH) model [36]:
   The RH model can be treated as a version of the LC model with an extra cohort component,

   $$ln(m_{xt}) = \beta_x^{(1)} + \beta_x^{(2)}\kappa_t^{(2)} + \beta_x^{(3)}\gamma_{t-x}^{(3)}, \tag{5}$$

   where $\sum_x \beta_x^{(2)} = 1, \sum_t \kappa_t^{(2)} = 0, \sum_x \beta_x^{(3)} = 1, \sum_{x,t}\gamma_{t-x}^{(3)} = 0$, and the parameter $\beta_x^{(i)}$ denotes the average age-specific mortality, $\kappa_t^{(2)}$ represents the general mortality level, and $\gamma_{t-x}^{(3)}$ reflects the cohort-related effect.

3. Cairns-Blake-Dowd (CBD) model [37]:
   The CBD model was designed to model mortality rates of older age groups and deal with longevity risk in pensions and annuities. The CBD model assumes that the mortality rates

satisfy

$$\text{logit}(m_{xt}) = \log\frac{m_{xt}}{1 - m_{xt}} = \beta_x^{(1)}\kappa_t^{(1)} + \beta_x^{(2)}\kappa_t^{(2)}, \tag{6}$$

where the parameters are $\beta_x^{(i)}$ and $\kappa_t^{(i)}$ $(i = 1, 2)$ denote the average age-specific mortality and general mortality levels. If we assume $\beta_x^{(1)} = 1$ and $\beta_x^{(2)} = x - \bar{x}$, then the model has a simple parametric form:

$$\text{logit}(m_{xt}) = \kappa_t^{(1)} + \kappa_t^{(2)}(x - \bar{x}). \tag{7}$$

4. The Age-Period-Cohort (APC) model:
The APC model is a popular tool for modelling disease incidence and mortality in epidemiology. Heuristically, if we consider the notion of Analysis of Variance, the LC model considers the effects of Age and Age×Period (interaction), whereas the APC model considers three main effects, Age, Period, and Cohort.

$$ln(m_{xt}) = \alpha_x + \kappa_t + \gamma_{t-x}, \tag{8}$$

with constraints $\sum_{c=t-x}\gamma_c = 0$ and $\sum_c c\gamma_c = 0$ [38].

## Section 3: Results

Taiwan's NHI has a concrete and rigorous, standard and review process for determining whether a person has CI. This helps the insurance industry prevent insurance disputes and develop CI-related products [15]. The CI products are among the most popular health products in Taiwan, and the experienced loss ratio of CI products meets expectations. As the size of exposures from the NHI database is fairly large, we expect that if the criteria used are reasonable, the prevalence, incidence, and mortality rates should be consistent and stable between ages and years, as well as satisfying experts' (such as doctors') opinions. Notably, we consider consistency and moral hazard to reduce insurance risk.

Note that the medical records in the NHIRD follow the International Classification of Diseases, 9th Revision (ICD-9); thus, we used the ICD code to determine whether people were diabetic. In particular, we were interested in T2DM, which accounts for 95% of diabetes cases in Taiwan (Source: Health Promotion Administration, Ministry of Health and Welfare). The ICD code of T2DM is 250, and the cases of type 1 diabetic (ICD code 250×1, 250×3) are excluded in this study. However, we did not rely solely on the ICD code to identify patients with diabetes, as it does not reveal the severity of diseases. We included other conditions, such as the number of outpatient visits, similar to the criteria for judging diabetes in S1 Table.

Prescription drugs are often included in the decision to treat diabetes. Unfortunately, there are concerns regarding the quality of prescription drug records, and according to our consultation with doctors, some people may even use diabetes prescription drugs for weight loss. Another reason for not using prescription drugs when deciding on treatment for diabetes is that patients may seek alternative treatments. Garrow [39] reported that 46.7% of patients with diabetes used complementary and alternative medicine. Additionally, it is difficult to develop a complete list of medicines for patients with diabetes. Thus, we sought another type of medical record for chronic diseases such as diabetes, called refillable (continuous) prescriptions for patients with chronic illness (RP). The RP has been enforced since 2003 and has significantly reduced the number of hospital visits.

Diabetes is usually not immediately fatal, therefore, patients often stop visiting doctors or forget to take regular medications when the symptoms of diabetes (such as hyperglycemia) are relieved. This would make it difficult to calculate the incidence rates of diabetes, for example, failure to identify first-time patients. Thus, we adapted rules similar to the idea of a washout period used in Taiwan's health insurance products. In Taiwan, usually a two-year observation (or probationary) period is used to reduce the possibility of moral hazard and overestimation. For example, if consumers want to purchase cancer insurance, they must provide their medical records over the last two years, showing that they have not yet had cancer. A two-year observation period was used to determine the incidence rate of diabetes.

With a two-year observation period, we calculated the incidence rates based on the number of outpatient visits and RP's. Logically, more outpatient visits should reduce the possibility of false positive results. Lin et al. [40] showed that the accuracy of the overall diabetes diagnosis in NHI claims data was 74.6%, which increased to 96.1% for cases with four or more outpatient visits. Using the criterion of four outpatient visits per year, we found that the prevalence rates of T2DM were stable in 2008–12 and were a reverse U-shaped curve, reaching a peak around age 80 (Fig 2). We also computed the prevalence rates of T2DM using the criteria of 2 and 3 outpatient visits per year, and they were higher than those of 4 outpatient visits per year; however, the results varied significantly for different years. The estimated results, based on four outpatient visits per year, were more consistent and the incidence rates were stable and followed a reverse U-shaped curve (left panel of Fig 3), reaching a peak of 2% around the age of 75. We also considered the incidence rates of T2DM using the criterion of one RP per year, and the results under this criterion were interesting (right panel of Fig 3). Interestingly, the incidence rates based on four outpatient visits per year and one RP visit per year were almost identical. As RP is easy to confirm, we used one RP per year to determine diabetes patients for the remainder of this study.

Fig 4 shows the age-specific mortality rates of patients with diabetes aged 71–84, compared with those of Taiwan's general population and Taiwan's cancer patients. As expected, the mortality rate in patients with diabetes was much lower than that in patients with cancer. However, the mortality rates of patients with diabetes were similar to those of Taiwan's general population; only female mortality rates were slightly higher. This result is somewhat different from

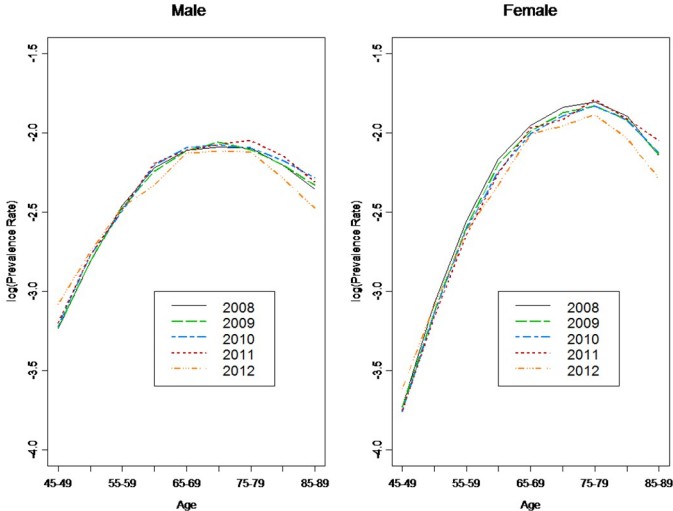

**Fig 2. Prevalence rates for four outpatient visits per year (2008–12).**

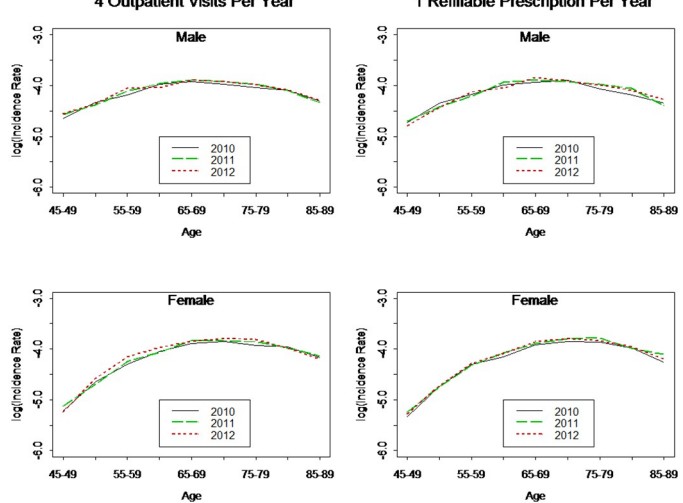

**Fig 3. Incidence rates of T2DM (2010–12).**

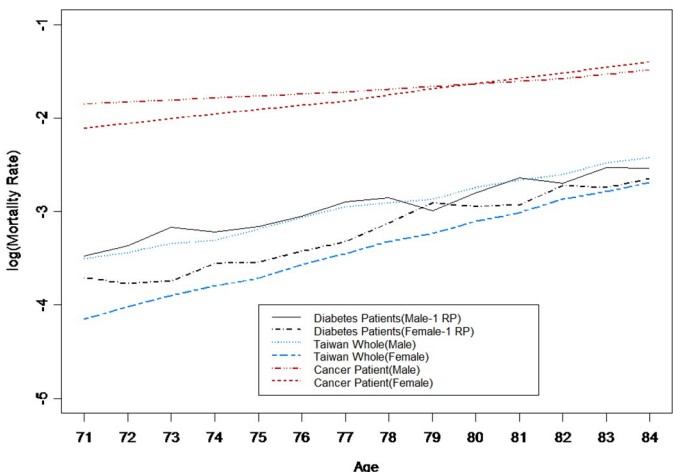

**Fig 4. Mortality rates of different populations.**

that of previous studies in which older patients with diabetes had higher mortality rates [29, 41, 42].

## Section 4: Discussion

In this study we used stochastic mortality models to estimate the incidence and mortality rates of diabetes and selected models with the smallest estimation errors (MAPE). We use these to design diabetes insurance products and discuss whether it is feasible to use commercial insurance products to deal with the challenges of prolonging life in Taiwan. Patients with diabetes were defined as those who have one RP per year. In addition, we considered the Generalized Age-Period-Cohort (GAPC) models using the R package StMoMo. If the population sizes were small, we combined graduation methods, such as the PSMR method, with mortality

models [43], using a combination of PSMR and LC models. We conducted the model evaluation with three data periods: 2005–2013, 2007–2012, and 2008–2013, in order to verify if the results of model fitting were time-dependent.

First, we present the results of the diabetes incidence rates. Two age groups were considered: single-age and 5-year age groups. For the single-age case, the age ranges are 45, 46, . . ., 89, while the age range are 45–49, 50–54, . . ., 95–99 for the 5-year age case; however, we do not consider ages 90 and beyond (i.e., 90+) for the single-age case because the population size of 90+ is too small. The results of 5-year age case (ages 45–99) are shown in Table 1. The APC model exhibits the best fit for all three periods. If we omit data from 2005 and 2006, the LC, APC, PSMR, and PSMR+LC models have satisfactory fitting results. The results of the single-age group (ages 45–89) are slightly different (Table 2), and RH has the smallest MAPE value, while the LC, APC, and PSMR+LC models have good fit.

As the LC model is frequently used in prediction, we used it to demonstrate the trend of diabetes incidence rates for the case of a 5-year age and years 2008–2013. We used the estimates of the LC model parameters to acquire the annual increment in incidence rates. In particular, we assume $\kappa_t^{(2)} = a + bt$ in Eq (4) and thus the annual increment of incidence rate at age $x$ is $\beta_x^{(2)} \times b$ [15]. In other words, the diabetes incidence rate of age $x$ at year $t+1$ is $e^{\beta_x^{(2)} \times b}$ times the diabetes incidence rate of age $x$ at year $t$. Fig 5 shows the annual increments of diabetes incidence rates at all age groups. The annual increments are smaller for younger groups and generally increase with age, reaching approximately 6% at ages 85–89. The scale of annual increments is worth noting. However, we need to collect more data for long-term projections because we have six years of data.

Modelling the mortality rates of patients with diabetes followed the same process. Owing to the nature of our data, we could only estimate the mortality rates for 2006–2011. This is

**Table 1. Fitting MAPE of incidence rates (ages 45–99, 5-age groups).**

|  | 2005–2013 | | 2007–2012 | | 2008–2013 | | Average |
|---|---|---|---|---|---|---|---|
|  | Male | Female | Male | Female | Male | Female |  |
| LC | 49.87 | 9.80 | 6.12 | 7.97 | 6.04 | 10.00 | 14.97 |
| APC | 6.40 | 6.71 | 4.49 | 7.03 | 4.07 | 4.89 | 5.60 |
| PSMR | 144.64 | 9.46 | 6.98 | 9.72 | 5.91 | 9.47 | 31.03 |
| PSMR+LC | 147.49 | 11.69 | 7.56 | 9.92 | 6.29 | 10.13 | 32.18 |
| CBD | 338.46 | 83.92 | 43.80 | 88.21 | 41.59 | 84.65 | 113.44 |
| RH | - - -[1] | 78.38 | 68.55 | 68.48 | 37.09 | 74.94 | 65.49 |

[1] In 2005, the incidence number of 95–99 was 0 and the RH model did not converge.

**Table 2. Fitting MAPE of incidence rates (ages 45–89, single-age).**

|  | 2005–2013 | | 2007–2012 | | 2008–2013 | | Average |
|---|---|---|---|---|---|---|---|
|  | Male | Female | Male | Female | Male | Female |  |
| LC | 10.37 | 10.16 | 7.80 | 7.22 | 8.22 | 7.65 | 8.57 |
| APC | 10.70 | 9.95 | 7.38 | 7.73 | 7.99 | 7.71 | 8.58 |
| PSMR | 10.87 | 10.34 | 8.51 | 8.39 | 8.77 | 8.24 | 9.19 |
| PSMR+LC | 13.21 | 12.46 | 8.95 | 8.66 | 8.95 | 8.62 | 10.14 |
| CBD | 30.24 | 40.40 | 26.34 | 38.43 | 25.87 | 37.23 | 33.08 |
| RH | 9.57 | 8.71 | 6.09 | 6.18 | 6.11 | 6.18 | 7.14 |

**Fig 5. Annual increments of diabetes incidence rates (LC model).**

because the death criteria were based on two-year washout period; thus, we could not estimate the mortality rates for 2012 and 2013. Nevertheless, we attempted to verify whether GAPC models can capture trends in mortality rates. However, owing to the consideration of sample size, the age ranges for the 5-year age and single-age groups were 70–74, 75–79, . . ., 95–99 and 70, 71, . . ., 89 years, respectively. Table 3 lists the fit errors with respect to MAPE for all models. Except for the RH model, all models had fairly accurate estimations, with an average MAPE of approximately 5%. We used the estimates of LC model parameters to acquire the annual increment of mortality rates for elderly diabetes patients, for the 5-year age group and years 2006–2011 (Fig 6). The annual increments were 3.6% and 1.6% for male and female patients, respectively. The annual increments were more stable for ages 70–89 but were reduced to 2.5% and 0.6% for male and female patients, respectively.

The results of the model evaluation of diabetes incidence and mortality rates suggest that the LC and APC models are preferred. The models indicated that the incidence and mortality rates increased with time; however, the increments in mortality rates (using the LC model) were much smaller. As a result, we expected that the number of patients with diabetes would increase over time, especially among the elderly. Patients with diabetes usually use more medical resources than those without diabetes; thus, more patients with diabetes indicate more medical expenditures for Taiwan's NHI. The Taiwanese government needs to look for

**Table 3. MAPE of mortality rates.**

| | 5-year age (70–99) | | Single-age (70–89) | | Average |
|---|---|---|---|---|---|
| | **Male** | **Female** | **Male** | **Female** | |
| **LC** | 3.76 | 2.74 | 5.48 | 4.75 | 4.18 |
| **APC** | 3.27 | 2.64 | 4.86 | 5.13 | 3.97 |
| **PSMR** | 4.60 | 4.41 | 6.37 | 5.75 | 5.28 |
| **PSMR+LC** | 4.94 | 4.41 | 6.42 | 5.75 | 5.38 |
| **CBD** | 5.56 | 4.41 | 6.59 | 5.95 | 5.63 |
| **RH** | 24.74 | 11.51 | 3.84 | 4.14 | 11.06 |

**Diabetes Mortality Rates**

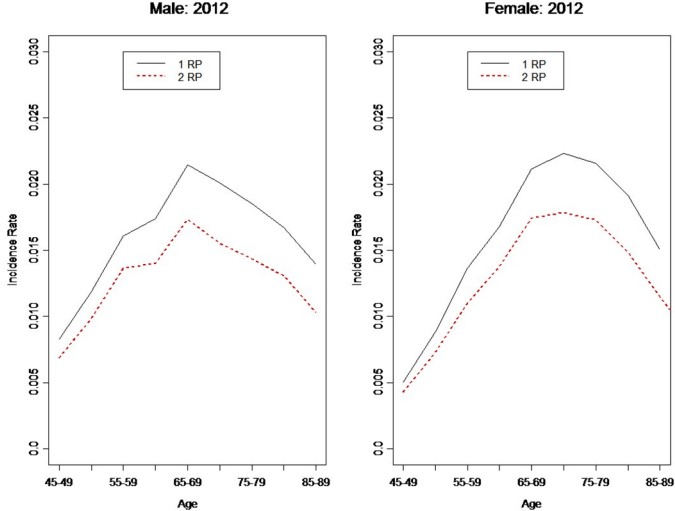

**Fig 6. Annual increments of diabetes mortality rates (LC model).**

solutions dealing with population aging and prolonging life to ensure the sustainability of the NHI and social insurance systems.

Additionally, the incidence rates (and possibly mortality rates) of diabetes depend on the judgment criteria. The trends in these rates may also differ significantly. We compared the results using 1 RP and 2 RP per year as the criteria. Fig 7 shows the incidence rates of those two criteria in 2012. Interestingly, for male and female patients, the diabetes incidence rates for 2 RP were approximately 20% lower than those of 1 RP. Diabetes-related mortality rates showed a similar pattern. As we could not compute the mortality rates for 2012, we compared the mortality rates for 2009. The mortality rates of patients with diabetes using 2 RP were approximately 7% lower than those using 1 RP (Fig 8). It appears that using RP can produce fairly stable estimates of incidence and mortality rates. Regarding the gaps between 1 RP and 2

**Fig 7. Diabetes incidence rates in Taiwan.**

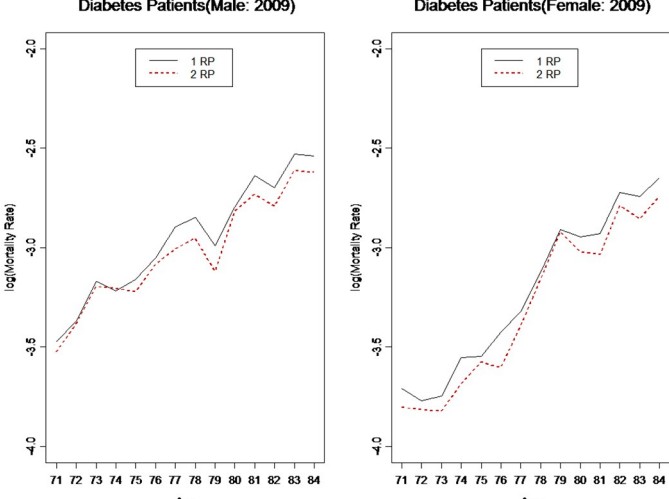

**Fig 8. Diabetes patients mortality rates.**

RP, we used methods such as the spill-over effect, similar to the car insurance no-claim discount method, to design medical policies with discounts for the insured who continue receiving treatment.

This study has three main limitations: the data period, definition of diabetes, and death criteria. We used only the datasets up until 2013 because there is a time gap in the release of diabetes data as Taiwan implemented the Personal Data Protection Act in 2012, increasing the difficulty of human-related research. Additionally, as mentioned in the Materials and Methods section, the definition of diabetes is controversial; thus, we used outpatient records to determine whether a person had diabetes. Our definition is related to the willingness to visit doctors to take medications regularly, and patients who do not visit doctors or take medications have higher mortality rates. Therefore, our results may not be applicable to other studies that use different definitions of diabetes. Similarly, the mortality rates of patients with diabetes were determined based on medical records, which may differ from official statistics.

## Section 5: Conclusions

Population aging is a common demographic phenomenon in the 21st century. As life expectancy increases, the proportion of the elderly population is expected to continue to increase. Chronic diseases such as metabolic syndrome and the replacement of infectious and acute diseases have become the main health concerns in many countries. Diabetes is a major metabolic syndrome; however, many people do not know whether they have diabetes. Unlike stroke and cardiovascular disease, diabetes has not received much attention. It has received increasing attention in recent years because previous studies have shown that diabetes is associated with many diseases. The 2016 annual report of the US Renal Registry [43] showed that the incidence and prevalence of kidney disease in Taiwan were the highest worldwide in 2014, with 455 and 3,219 people per million people per year, and 45% of dialysis patients were diagnosed with diabetes. Diabetes will have a larger influence on the health and medical expenditure of Taiwanese people; thus, we used Taiwan's National Health Insurance Research Database to explore trends in T2DM. In this study, we evaluated the criteria for diabetes and calculated its

incidence and mortality rates based on NHI records as it covered approximately 99.9% of Taiwanese citizens by the end of 2023.

Using RP, we obtained stable incidence and mortality rates that could be used to design diabetes insurance products. Our results show that when patients with diabetes continue to receive treatment, their mortality rates are not significantly different from those of the general population. This discovery can be regarded as an application of big data that provides new insights for insurance companies in product design, and provides policyholders with more opportunities to purchase insurance products. In addition, among all GAPC models for fitting the incidence and mortality rates, the APC model had the smallest MAPE errors, and the LC model was a feasible choice. When we used the LC model to measure the time trend, we found that the incidence rates of T2DM gradually increased with time, whereas the mortality rates of elderly patients with diabetes changed with a stable path.

An aging population and unhealthy lifestyle can lead to changes in the main causes of death in many countries, such as Taiwan [44]. Diabetes appears to be a good indicator for the degree of unhealthy level. According to the National Diabetes Statistics Report (2020), complications in adults in the U.S. (aged 18 and above) diagnosed with diabetes in 2013–2016 included overweight and obesity, physical inactivity, high blood pressure, high cholesterol, and high blood glucose, which are related to metabolic syndrome diseases. Thus, the increasing incidence of diabetes in Taiwan and the U.S. indicates increasing medical demands and expenditures, not restricted to the number of deaths. In order to maintain the sustainability of the NHI, we suggest that Taiwan's government provide more incentives for diabetes patients to pay extra attention to their health, such as free health examinations every two or three years.

For commercial insurance, diabetes can be considered a sign of potential health problems; thus, we treated it as a risk factor (i.e., those with diabetes as part of the sub-standard group) for insurance products. However, a health exam is usually not required for commercial insurance in Taiwan, and it is difficult to verify whether the insured have diabetes, similar to verifying whether they are over-weight or use tobacco regularly. Alternatively, the concept of insurance product options can be adopted when designing diabetes products. For example, consumers can purchase options to treat diabetes. When the insured are diagnosed with diabetes, instead of receiving a benefit payment, they can purchase new policies at the standard price rate. This is feasible for life insurance products because the mortality rates of patients with diabetes can be determined. For health insurance products, further studies and more information regarding the relationship between diabetes and other health conditions are needed.

## Supporting information

**S1 Table. Disease definition of diabetes in the past studies.**
(TIFF)

## Author Contributions

**Data curation:** Jack C. Yue, Hsin-Chung Wang, Ting-Chung Chang.

**Formal analysis:** Ting-Chung Chang.

**Investigation:** Jack C. Yue, Hsin-Chung Wang, Ting-Chung Chang.

**Methodology:** Jack C. Yue, Hsin-Chung Wang.

**Software:** Hsin-Chung Wang.

**Visualization:** Hsin-Chung Wang, Ting-Chung Chang.

**Writing – original draft:** Jack C. Yue.

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
