## [Decision Letter · Decision Letter 0]

9 Feb 2024

PONE-D-23-30690Application of Type II Diabetes Incidence and Mortality Rates for InsurancePLOS ONE

Dear Dr. Wang,

Thank you for submitting your manuscript to PLOS ONE. After careful consideration, we feel that it has merit but does not fully meet PLOS ONE’s publication criteria as it currently stands. Therefore, we invite you to submit a revised version of the manuscript that addresses the points raised during the review process.

Dear authors, Please find the external reviewers' commmets below. We would like to suggest that you revise the manuscript and submit a major revision to us at your earliest convenience.

We look forward to receiving your revised manuscript.

Kind regards,

Shuo-Yan Gau

Academic Editor

PLOS ONE

Journal Requirements:

4. We notice that your supplementary tables are included in the manuscript file. Please remove them and upload them with the file type 'Supporting Information'. Please ensure that each Supporting Information file has a legend listed in the manuscript after the references list.

Reviewers' comments:

Reviewer's Responses to Questions

**Comments to the Author**

1. Is the manuscript technically sound, and do the data support the conclusions?

Reviewer #1: Partly

Reviewer #2: Partly

2. Has the statistical analysis been performed appropriately and rigorously? 

Reviewer #1: Yes

Reviewer #2: I Don't Know

3. Have the authors made all data underlying the findings in their manuscript fully available?

Reviewer #1: Yes

Reviewer #2: Yes

4. Is the manuscript presented in an intelligible fashion and written in standard English?

Reviewer #1: Yes

Reviewer #2: Yes

5. Review Comments to the Author

Reviewer #1: The authors present an analysis of type 2 diabetes incidence and mortality rates in Taiwan using data from the National Health Insurance Research Database (NHIRD). The goals are to model diabetes trends over time and demonstrate applications for insurance product design. The topic is interesting and important given the growing burden of diabetes globally. However, the paper needs some improvement in terms of structure, clarity, and discussion.

Major comments:

1. The structure of the paper should follow the conventional order of Materials and Methods, Results, Discussion, and Conclusions. This would make the paper more coherent and easier to follow. Currently, the paper seems to mix the discussion with other sections, which reduces the impact of the main findings.

2. The introduction should be more concise and focused on the rationale and objectives of this specific study. There is too much background information on general trends in population aging, chronic disease, and diabetes prevalence that may not be directly relevant to the research question. The introduction should clearly state the research gap, the hypothesis, and the contribution of the paper.

3. Generally, the context should also be more concise and contain only relevant information to this study. For example, lines 263-268 use too many sentences/words to introduce RP and its benefits, which are irrelevant to this study.

4. The discussion should elaborate more on the implications of the findings and limitations. For example, how could the data inform specific insurance product designs or public health policies? What are the potential benefits and challenges of using such data for insurance purposes? What are the limitations of claims data for identifying diabetes cases and measuring outcomes? How do the results compare with other studies in the literature?

Minor comments:

1. Use abbreviations consistently, e.g. T2DM vs. type 2 diabetes. Define the abbreviations at the first occurrence and use them throughout the paper.

2. In line 175, this is the first time “SMR” shows up, please define it there.

3. In lines 181-183, does SMR being "larger" than 1 indicate higher mortality rates?

4. In lines 288-289, please supplement the results using criteria 2 and 3 outpatient visits per year as a reference.

5. In line 325, please define PSMR.

6. In lines 334-335, please provide the result when you omit the data in 2005 and 2006 as supplementary data. What is the reason why omitting the data in these two years would result in satisfactory fitting results?

The paper has a sound analysis approach and a useful contribution, but it needs some revisions to address the major comments above. The results and implications could be clarified and expanded on further. With the revision, this could become a strong paper for publication.

Reviewer #2: 1) The use of "metabolic syndrome" in the abstract (lines 31-32) requires clarification. This term is typically defined as a group of conditions increasing the risk of heart disease, stroke, and diabetes, and should not be conflated with these diseases themselves. People who have heart disease, stroke, or diabetes do not necessarily have metabolic syndrome.

2) Please list the limitations of your study in the discussion section.

3) The data sets you used only cover up until 2013, and there have been significant advancements in diabetes treatment since that time.

4) The implications of your findings for insurance product design are interesting. Further discussion could explore how these findings might practically influence current insurance models and their implications for policyholders and insurance companies

6. PLOS authors have the option to publish the peer review history of their article (what does this mean?). If published, this will include your full peer review and any attached files.

Reviewer #1: **Yes: **Shao-Wei Lo

Reviewer #2: No

---

## [Author Response · Author response to Decision Letter 0]

7 Apr 2024

We greatly appreciate the insightful comments from the editor and two anonymous reviewers, which helped us to clarify the context of our work. Please see the Communications_T2DM for Insurance (20240406).docx file.

Dear Editor,

This is in reply to your letter of Feb. 10, 2024 regarding our manuscript, Application of Type II Diabetes Incidence and Mortality Rates for Insurance. We appreciate the helpful and insightful comments from the anonymous referees. We have revised the manuscript substantially in light of these comments (marked in red), and hereby enclose a copy of the revised manuscript. 

Reviewer #1

Major Comments

1. The structure of the paper should follow the conventional order of Materials and Methods, Results, Discussion, and Conclusions. This would make the paper more coherent and easier to follow. Currently, the paper seems to mix the discussion with other sections, which reduces the impact of the main findings.

Response: 

We have revised the order of our manuscript according to the reviewer’s suggestion, and the manuscript follows the order of Materials and Methods, Results, Discussion, and Conclusions. 

2. The introduction should be more concise and focused on the rationale and objectives of this specific study. There is too much background information on general trends in population aging, chronic disease, and diabetes prevalence that may not be directly relevant to the research question. The introduction should clearly state the research gap, the hypothesis, and the contribution of the paper.

Response: 

We appreciate the reviewer’s comment and thus have revised the introduction, making it more concise and focused. In addition to removing unnecessary background information, we state the research gap, the hypothesis, and the contribution of the paper in the penultimate paragraph of the introduction. 

3. Generally, the context should also be more concise and contain only relevant information to this study. For example, lines 263-268 use too many sentences/words to introduce RP and its benefits, which are irrelevant to this study. 

Response: 

We have revised the manuscript according to the reviewer’s suggestion, making the paper more concise and containing only relevant information. 

4. The discussion should elaborate more on the implications of the findings and limitations. For example, how could the data inform specific insurance product designs or public health policies? What are the potential benefits and challenges of using such data for insurance purposes? What are the limitations of claims data for identifying diabetes cases and measuring outcomes? How do the results compare with other studies in the literature?

Response: 

We appreciate the reviewer’s comments. This study has three main limitations, including the data period, the definition of diabetes, and the death criteria. We use only the data sets up until 2013, since there is a time gap in the release of diabetes data and Taiwan implemented the Personal Data Protection Act in 2012, increasing the difficulty of human-related research. Also, as mentioned in the materials and methods section, the definition of diabetes is controversial and thus we use the outpatient records to judge whether a person has diabetes. Our study results may not be applied to other studies if different diabetes criteria are sued. Similarly, the mortality rates of diabetes patients are determined based on medical records, which may be different from the official statistics or other studies. For example, previous studies showed that older diabetes patients have higher mortality rates. We have added the preceding description in the discussion section.

As for the implication of this study, Diabetes is generally thought to be associated with a higher mortality rate in Taiwan and diabetes patients are not easy to purchase life insurance products. Our analysis results show that if patients with diabetes continue to receive treatment, their mortality rates are not much different from those of the general population. In other words, we applied the idea of big data to verify that people with diabetes do not necessarily live shorter. It provides new thinking for insurance companies in product design, as well as giving policyholders more opportunities to purchase insurance products. 

Minor Comments

1. Use abbreviations consistently, e.g. T2DM vs. type 2 diabetes. Define the abbreviations at the first occurrence and use them throughout the paper.

Response: 

We have defined the abbreviation at the first occurrence and use them throughout the manuscript. 

2. In line 175, this is the first time “SMR” shows up, please define it there.

Response: 

We have added the definition of SMR when it was first mentioned.

3. In lines 181-183, does SMR being “larger” than 1 indicate higher mortality rates?

Response: 

We apologize for the error and have corrected it. 

4. In lines 288-289, please supplement the results using criteria 2 and 3 outpatient visits per year as a reference.

Response: 

We appreciate the reviewer’s suggestion. In this article, we mainly focus on the rates estimated under the criteria of 4 outpatient visits per year, but we still calculate those under other criteria. We found that the diabetes prevalence rates under criteria 2 and 3 outpatient visits per year results are slightly larger than those under 4 outpatient visits per year. For example, Figure A-1 shows the diabetes prevalence rates under 2 and 4 outpatient visits per year. 

Fig A-1. Diabetes Prevalence Rates of 2 and 4 Outpatient Visits per Year (2008-12)

5. In line 325, please define PSMR.

Response: 

We appreciate the reviewer’s comment and we have added the definition of PSMR in lines 173-178.

6. In lines 334-335, please provide the result when you omit the data in 2005 and 2006 as supplementary data. What is the reason why omitting the data in these two years would result in satisfactory fitting results?

Response: 

We appreciate the reviewer’s suggestion. We propose an observation period of two years for determining if a person had diabetes to avoid moral hazard. The analysis results (Table 2) support our idea and omitting the years 2005 and 2006 would result in more stable diabetes incidence rates. In addition, the following graph (Fig A.2) show the male age-specific incidence rates of diabetes, 2005~2012, and the results of 2005 and 2006 look different than those of other years. Due to space limitations, this figure is not included in the article.

Fig A-2. Estimated Age-specific Incidence Rates of Diabetes (Male, 2005~2012)

Reviewer #2

Major Comments

1. The use of “metabolic syndrome” in the abstract (lines 31-32) requires clarification. This term is typically defined as a group of conditions increasing the risk of heart disease, stroke, and diabetes, and should not be conflated with these diseases themselves. People who have heart disease, stroke, or diabetes do not necessarily have metabolic syndrome.

Response: 

We appreciate the reviewer’s suggestion and have revised the abstract. 

2. Please list the limitations of your study in the discussion section.

Response: 

We appreciate the reviewer’s suggestion regarding the limitations of our study. This study has three main limitations, including the data period, the definition of diabetes, and the death criteria. We use only the data sets up until 2013, since there is a time gap in the release of diabetes data and Taiwan implemented the Personal Data Protection Act in 2012, increasing the difficulty of human-related research. Also, as mentioned in the materials and methods section, the definition of diabetes is controversial and thus we use the outpatient records to judge whether a person has diabetes. Our study results may not be applied to other studies if different diabetes criteria are sued. Similarly, the mortality rates of diabetes patients are determined based on medical records, which may be different from the official statistics. We have added the preceding description in the discussion section. 

3. The data sets you used only cover up until 2013, and there have been significant advancements in diabetes treatment since that time.

Response: 

We appreciate the reviewer’s comments regarding the data used. There is a time gap in the release of Taiwan’s diabetes data, including National Health Insurance data. For example, Tsai and Li (2023) explored whether daily diet is related to the risk of type 2 diabetes in Taiwan and they used 2013-2016 data. Annals of Diabetes Mellitus in Taiwan 2019 Update on Type 2 Diabetes, published by Diabetes Association of the Republic of China (Taiwan), were based on the data from 2014 and before. Also, Taiwan implemented the Personal Data Protection Act in 2012, and there have been more and more restrictions on data application and use, making human-related research more difficult. As for the issue of diabetes treatment, diabetes has been one of the top ten causes of death in Taiwan for more than 30 years, and about 6% of annual deaths are diabetes related in past 15 years. Diabetes continues to poses a serious threat to Taiwan’s people, and treatments still need to be developed.

Tsai TJ and Li MC. Adherence to the Taiwan Daily Food Guide and the Risk of Type 2 Diabetes: A Populational Study in Taiwan. International Journal of Environmental Research and Public Health. 2023 Feb; 20(3): 2246.

4. The implications of your findings for insurance product design are interesting. Further discussion could explore how these findings might practically influence current insurance models and their implications for policyholders and insurance companies.

Response: 

 We appreciate the reviewer’s suggestion and we have included a discussion of possible impacts of our findings to Taiwan’s insurance industry. Diabetes is generally thought to be associated with a higher mortality rate in Taiwan and the persons with diabetes are not easy to purchase life insurance. Our analysis results show that if patients with diabetes continue to receive treatment, their mortality rates are not much different from those of the general population. This discovery can be regarded as an application of big data. In addition, it provides new thinking for insurance companies in product design and gives policyholders more opportunities to purchase insurance products. We have added the preceding description in the conclusion section.

---

## [Decision Letter · Decision Letter 1]

8 May 2024

PONE-D-23-30690R1Application of Type II Diabetes Incidence and Mortality Rates for InsurancePLOS ONE

Dear Dr. Wang,

Thank you for submitting your manuscript to PLOS ONE. After careful consideration, we feel that it has merit but does not fully meet PLOS ONE’s publication criteria as it currently stands. Therefore, we invite you to submit a revised version of the manuscript that addresses the points raised during the review process.

**ACADEMIC EDITOR: **Dear authors, We appreciate your response to the comments raised by the reivewers. Your study has been favorably reviewed by our expert referees, who found your paper to be of interest. However, there were some points at issue that deserve your attention and revisions must be made according to the reviewers' comments.

We look forward to receiving your revised manuscript.

Kind regards,

Shuo-Yan Gau

Academic Editor

PLOS ONE

Reviewers' comments:

Reviewer's Responses to Questions

**Comments to the Author**

1. If the authors have adequately addressed your comments raised in a previous round of review and you feel that this manuscript is now acceptable for publication, you may indicate that here to bypass the “Comments to the Author” section, enter your conflict of interest statement in the “Confidential to Editor” section, and submit your "Accept" recommendation.

Reviewer #1: (No Response)

Reviewer #2: All comments have been addressed

2. Is the manuscript technically sound, and do the data support the conclusions?

Reviewer #1: Partly

Reviewer #2: (No Response)

3. Has the statistical analysis been performed appropriately and rigorously? 

Reviewer #1: I Don't Know

Reviewer #2: (No Response)

4. Have the authors made all data underlying the findings in their manuscript fully available?

Reviewer #1: Yes

Reviewer #2: (No Response)

5. Is the manuscript presented in an intelligible fashion and written in standard English?

Reviewer #1: No

Reviewer #2: (No Response)

6. Review Comments to the Author

Reviewer #1: 

Dear Author,

Thank you for submitting your manuscript. I appreciate the effort you’ve put into your work, and I believe it has the potential to contribute significantly to the field. However, I have a few suggestions to enhance the clarity and impact of your paper.

# Language and Clarity:

Consider hiring an official English editor to improve the language quality. While your manuscript is generally well-written, a professional review could further enhance its readability. For example,

- In the abstract, references are generally not necessary. For instance, please review lines 35 and 39.

- In line 51, the hyphen in “Be-coming” is redundant. You can simply use “Becoming.”

- Regarding line 57, you can choose either “people ages 65 and over” or simply “elderly.” The parentheses provide no additional explanation.

- Throughout the paper, there is an overuse of the future tense (“will”). Please ensure consistency with the actual actions performed during your study to improve overall composition.

#Structural Improvements:

I commend your efforts to make the manuscript well-structured and concise. However, there are instances where statements seem misplaced:

- In the “Materials and Methods” section (lines 105-108), you mention that the National Health Insurance Research Database (NHIRD) covers about 99.6% of Taiwan’s citizens. While this information underscores the strength of your study, it might be more appropriate for the discussion section.

- Similarly, lines 129-136 discuss the challenge of determining appropriate criteria for designing insurance products. This context could be better placed in the introduction, where you introduce the problem you aim to address.

- Be mindful of similar issues throughout the article and in the results section.

Reviewer #2: (No Response)

7. PLOS authors have the option to publish the peer review history of their article (what does this mean?). If published, this will include your full peer review and any attached files.

Reviewer #1: No

Reviewer #2: No

---

## [Author Response · Author response to Decision Letter 1]

8 Jun 2024

Reviewer #1

Major Comments

1. Consider hiring an official English editor to improve the language quality. While your manuscript is generally well-written, a professional review could further enhance its readability. For example, 

- In the abstract, references are generally not necessary. For instance, please review lines 35 and 39.

- In line 51, the hyphen in “Be-coming” is redundant. You can simply use “Becoming.”

- Regarding line 57, you can choose either “people ages 65 and over” or simply “elderly.” The parentheses provide no additional explanation.

- Throughout the paper, there is an overuse of the future tense (“will”). Please ensure consistency with the actual actions performed during your study to improve overall composition.

Response: 

The previous version of our manuscript was proofread by a professional English editor, however, we hired another English editor to review the revised manuscript. Additionally, we have removed the references in the abstract and revised the incorrect words, such as “elderly” and “will.” 

2. I commend your efforts to make the manuscript well-structured and concise. However, there are instances where statements seem misplaced:

- In the “Materials and Methods” section (lines 105-108), you mention that the National Health Insurance Research Database (NHIRD) covers about 99.6% of Taiwan’s citizens. While this information underscores the strength of your study, it might be more appropriate for the discussion section.

Response: 

 We have moved the NHI enrollment information to the discussion section. In addition, the enrollment figure has been updated to 99.9% according 2023 data. 

- Similarly, lines 129-136 discuss the challenge of determining appropriate criteria for designing insurance products. This context could be better placed in the introduction, where you introduce the problem you aim to address.

Response: 

 We have moved the text on challenge of determining appropriate criteria for designing insurance products to the Introduction section.

- Be mindful of similar issues throughout the article and in the results section.The structure of the paper should follow the conventional order of Materials and Methods, Results, Discussion, and Conclusions. This would make the paper more coherent and easier to follow. Currently, the paper seems to mix the discussion with other sections, which reduces the impact of the main findings.

Response: 

We have revised the manuscript to ensure that the discussion is not mixed with other sections.

---

## [Decision Letter · Decision Letter 2]

8 Jul 2024

Application of Type II Diabetes Incidence and Mortality Rates for Insurance

PONE-D-23-30690R2

Dear Dr. Wang,

We’re pleased to inform you that your manuscript has been judged scientifically suitable for publication and will be formally accepted for publication once it meets all outstanding technical requirements.

Kind regards,

Shuo-Yan Gau

Academic Editor

PLOS ONE

Additional Editor Comments (optional):

Reviewers' comments:

Reviewer's Responses to Questions

**Comments to the Author**

1. If the authors have adequately addressed your comments raised in a previous round of review and you feel that this manuscript is now acceptable for publication, you may indicate that here to bypass the “Comments to the Author” section, enter your conflict of interest statement in the “Confidential to Editor” section, and submit your "Accept" recommendation.

Reviewer #1: All comments have been addressed

Reviewer #2: All comments have been addressed

2. Is the manuscript technically sound, and do the data support the conclusions?

Reviewer #1: (No Response)

Reviewer #2: (No Response)

3. Has the statistical analysis been performed appropriately and rigorously? 

Reviewer #1: (No Response)

Reviewer #2: (No Response)

4. Have the authors made all data underlying the findings in their manuscript fully available?

Reviewer #1: (No Response)

Reviewer #2: (No Response)

5. Is the manuscript presented in an intelligible fashion and written in standard English?

Reviewer #1: (No Response)

Reviewer #2: (No Response)

6. Review Comments to the Author

Reviewer #1: (No Response)

Reviewer #2: (No Response)

7. PLOS authors have the option to publish the peer review history of their article (what does this mean?). If published, this will include your full peer review and any attached files.

Reviewer #1: No

Reviewer #2: No

---

## [Editor Report · Acceptance letter]

17 Jul 2024

PONE-D-23-30690R2 

PLOS ONE

Dear Dr. Wang, 

I'm pleased to inform you that your manuscript has been deemed suitable for publication in PLOS ONE. Congratulations! Your manuscript is now being handed over to our production team.

Kind regards, 

on behalf of

Dr. Shuo-Yan Gau 

Academic Editor

PLOS ONE